# 3D-printed Surgical Training Model Based on Real Patient Situations for Dental Education

**DOI:** 10.3390/ijerph17082901

**Published:** 2020-04-22

**Authors:** Marcel Hanisch, Elke Kroeger, Markus Dekiff, Maximilian Timme, Johannes Kleinheinz, Dieter Dirksen

**Affiliations:** 1Department of Cranio-Maxillofacial Surgery, University Hospital Münster, Albert-Schweitzer-Campus 1, Building W 30, D-48149 Münster, Germany; maximilian.timme@ukmuenster.de (M.T.); johannes.kleinheinz@ukmuenster.de (J.K.); 2Department of Cranio-Maxillofacial Surgery, Klinikum Osnabrück, Am Finkenhügel 1, 49076 Osnabrück, Germany; kroeger.elke@gmx.de; 3Department of Prosthetic Dentistry and Biomaterials, University Hospital Münster, Albert-Schweitzer Campus 1, D-48149 Münster, Germany; markus.dekiff@uni-muenster.de (M.D.); dirksdi@uni-muenster.de (D.D.)

**Keywords:** 3D printing, surgical training model, 3D rapid prototyping, root resection, CAD/CAM, dental education

## Abstract

*Background:* Most simulation models used at university dental clinics are typodonts. Usually, models show idealized eugnathic situations, which are rarely encountered in everyday practice. The aim of this study was to use 3D printing technology to manufacture individualized surgical training models for root tip resection (apicoectomy) on the basis of real patient data and to compare their suitability for dental education against a commercial typodont model. *Methods:* The training model was designed using CAD/CAM (computer-aided design/computer-aided manufacturing) technology. The printer used to manufacture the models employed the PolyJet technique. Dental students, about one year before their final examinations, acted as test persons and evaluated the simulation models on a visual analogue scale (VAS) with four questions (Q1–Q4). *Results:* A training model for root tip resection was constructed and printed employing two different materials (hard and soft) to differentiate anatomical structures within the model. The exercise was rated by 35 participants for the typodont model and 33 students for the 3D-printed model. Wilcoxon rank sum tests were carried out to identify differences in the assessments of the two model types. The alternative hypothesis for each test was: “The rating for the typodont model is higher than that for the 3D-printed model”. As the p-values reveal, the alternative hypothesis has to be rejected in all cases. For both models, the gingiva mask was criticized. *Conclusions:* Individual 3D-printed surgical training models based on real patient data offer a realistic alternative to industrially manufactured typodont models. However, there is still room for improvement with respect to the gingiva mask for learning surgical incision and flap formation.

## 1. Introduction

Digital technology has entered the dental practice on a broad front, as was predicted in the past [1]. By now, certain dental works can be performed in a completely digital workflow [2,3]. Consequently, it was only a matter of time before the technologies were transferred to dental education. A special focus here is on 3D printing alias additive manufacturing (AM) [4]. 3D printing converts virtual 3D models from computer-aided design (CAD) into haptic objects [5]. In recent years, the development of 3D printing technologies for medical and dental applications has increased significantly. The driving force behind the advancement of 3D printing for these applications is the possibility of creating personalized products, savings on small production runs, easier sharing and easier processing of patient image data [6]. It is expected that these properties of 3D printing will also have great advantages for dental education [6].

Since in dentistry, as with surgery, practical skills are particularly important, the use of training models has been established for years. These models are designed to teach the future dentist practical skills until he or she can apply them to real patients. In dental education, examination situations must also be considered. To maintain fairness, standardized models for conducting examinations should be implemented. Moreover, models can also be used to visualize special anatomical or pathological situations. This means that (individual) models are of particular importance in dental education, even outside the field of practical training. Last but not least, the models should also promote the students’ enthusiasm for education. Good, varied models are therefore a lever for the training of motivated and competent dentists.

In addition to the utilization of haptic models for the training of dentists, the use of virtual reality techniques in training is currently being discussed [7]. However, Nassar and Tekian in (2020) demonstrated in a recent review that an uncritical use of these new technologies is not very promising [8]. The authors of the present study see a great advantage of haptic models in the possibility of learning proprioceptive skills. This aspect has already been emphasized by other authors in the past [9]. 

Most simulation models used at university dental clinics are typodonts from industrial manufacturers. These models are offered with replaceable teeth and a replaceable gingiva mask, and usually show idealized eugnathic situations, which do not adequately reflect real situations encountered in everyday practice. Furthermore, the ready-made standard models do not usually depict special pathological or anatomic situations. Hence, several authors have developed and presented individualized training models in recent years; for example, Güth et al. fabricated a typodont implant simulation model in 2010 and integrated it into dental education [10]. This model was developed in a university and industry collaboration and is now commercially available. 

Lambrecht et al. developed surgical training models in 2010 based on real patient data and fabricated them by 3D printing [11]. Since then, rapid advances in 3D printing technologies have allowed more and more educators to produce their own training models [6,12]. In 2017, our work group presented a 3D-printed training model focusing on conservative and prosthetic procedures [13]. 

Werz et al. evaluated an inexpensive 3D printing technology (Fused Deposition Modeling /FDM) for producing individualized surgical models for training an external maxillary sinus lift and the extraction of impacted mandibular third molars [14]. They highlighted the possibility of producing models in small quantities and thus the ability to constantly vary the training scenarios.

Marty et al. presented a comparison between students’ perceptions of 3D-printed and series models in paediatric dentistry [15]. 

In a recent study from 2019, Höhne and colleagues [16] used 3D printing to create tooth models in which enamel and dentin could be distinguished. This set these printed models apart from conventionally fabricated models. The learning effect of preparing a crown was rated as good by the students in this study [16]. Shortly before, Höhne and Schmitter had produced 3D-printed models with carious lesions and had them evaluated by students. These models also received a good rating [17].

In summary, there are very promising approaches for the use of 3D-printed models in dental education. It is therefore reasonable to assume that this approach should be further pursued and scientifically investigated. 

In this contribution, we present an individualized 3D-printed surgical training model for root tip resection (apicoectomy), i.e., the removal of an inflammation around the root tip.

In the typodont model used by the students at our university to practice root tip resection (A-J OP OK, Frasaco^®^, Tettnang, Germany), the teeth are in direct contact with the hard plastic that simulates the jawbone. A sensitive periodontal ligament is not present. The apical granuloma, i.e., an inflammation at the root tip, in turn is simulated in wax. The teeth used are idealized stereotypes. Anatomical variations, such as extremely long or even curved roots, cannot be simulated with these industrially produced models. Therefore, we have developed a method to create more realistic, individualized training models.

We describe in detail the construction of the new 3D-printed surgical training model, which is based on a real patient. It depicts an upper jaw with three anterior root apices, the periodontal ligament around them and an apical granuloma. The material used to represent the periodontal ligament and the apical granuloma is softer than the material used for the other parts of the model. This allows a more realistic representation than in the typodont model. The new model also features a silicone gingival mask for practicing the surgical incision. The mask is manually fabricated directly on the model with a 3D-printed matrix. We also present an evaluation of the model by dental students and compare it with their evaluation of the conventional typodont model. Our intention was to evaluate whether dental students accept the 3D-printed surgical training model just as well as the popular typodont model.

## 2. Methods

### 2.1. Design

The training model was designed on the basis of a real patient situation with added inflamed tissue using CAD/CAM technology. A conventional impression of the real patient situation was made for the fabrication of a plaster cast. In order to make room for the gingival mask and to imitate the buccal bone lamella, the buccal surface of the anterior maxillary region was trimmed with rotating instruments. The gingiva was then modelled with a layer of wax 1 mm thick (Figure 1). In the next step, the shape of the modified plaster cast was acquired with an industrial 3D scanner (Atos I Rev02, GOM, Braunschweig, Germany) with and without the wax gingival mask. Each scan yielded a 3D model in form of a triangle mesh. In addition, 3D models/meshes of teeth 11, 12 and 21 (Figure 2) were created from the cone beam computed tomography (CBCT) data of another patient using Mimics Innovation Suite 19 software (Materialise, Leuven, Belgium) for the 3D reconstruction and Rhinoceros 5 (McNeel Europe, Barcelona, Spain) for modifications. 

The upper parts of the teeth’s meshes (crown, upper part of the tooth root) were deleted. The remaining parts were thickened by 0.25 mm in Geomagic Wrap (3D Systems, Rock Hill, SC, USA) to create a model of the periodontal ligament (Figure 3), which is missing in the typodont model. 

To simulate an apical granuloma on tooth 11, a sphere with a diameter of 6 mm surrounding the root apex of tooth 11 was constructed in Rhinoceros 5. The surface normals of the meshes representing the granuloma and the periodontal ligaments were inverted (in Rhinoceros 5, Figure 4) in such a way that the printing software treated these areas as cavities that were automatically filled with a soft support material during the 3D print. To mount the model on a standard phantom head, a baseplate and a reusable socket were designed in Rhinoceros 5. After adding the baseplate to the mesh of the modified plaster cast without wax layer, the resulting mesh was exported together with the meshes of the periodontal ligament and the granuloma into a single STL file for printing (Figure 4). 

For economic reasons, the model can be divided into multiple sections, which can be printed separately. 

The gingival mask on the 3D-printed model was manufactured using a matrix technique. The 3D model of the matrix was designed using the 3D models of the maxilla plaster cast, with and without wax layer. For the fabrication of the gingival mask, silicone (GI-MASK, Coltene^®^, Liechtenstein) was injected into the recess of the matrix and pressed against the 3D-printed training model (Figure 5). 

### 2.2. Printing

The printer used for manufacturing the models (Objet Eden 260V, Stratasys, Rehovot, Israel) employed the PolyJet technique. With this technique, models were built up sequentially from thin layers of liquid photopolymer (MED690, Stratasys, Rehovot, Israel; tensile strength: 54–65 MPa) that were cured with UV light after each pass. Undercut areas were supported with a softer material (SUP705, Stratasys) that could be removed with a water jet. Cavities intentionally placed inside the model to simulate periodontal ligament and apical granuloma were also filled with this soft support material. Up to 12 models were printed in one run, which took about 6 h. 

### 2.3. Evaluation of the 3D-printed Surgical Training Model 

Students of dentistry in their 9th semester at our university clinic, about one year before their final examinations, acted as test persons and evaluated the simulation models within the framework of the clinical “Surgical Course I” during the summer semester 2016 and the winter semester 2016/17. Informed consent from the participants was obtained.

The course was divided into two groups across two semesters, wherein group 1 had to evaluate the industrially manufactured training model (Frasaco^®^, Tettnang, Germany) and group 2 the new 3D-printed simulation model. Each participant carried out the root tip resection exercise only once to prevent biased results through a learning effect.

All of the participants received an iPad (Apple^®^, Cupertino, CA, USA) with a video showing the exercise. The exercise was not graded.

After the exercise, they were asked about their opinions regarding the concept and execution of the simulation model, as well as the difficulty of the exercise. For this purpose, a questionnaire was handed out, where the agreement to each statement could be rated on a visual analogue scale (VAS). For evaluation of the pseudonymous data, the ratings were mapped into an interval (0–10). The questionnaire also included an optional free-text section. For the statistical analysis, the software R (The R Project for Statistical Computing, www.R-project.org) was used.

Prior to the study, the sample size was calculated to be 29 participants per group, whereby a difference of 0.5 points was considered statistically significant. The sample size calculation and statistical set-up was done by Dr. Raphael Koch (Institute of Biostatistics and Clinical Research, University of Münster). Statistical analysis was done by D.D. The primary endpoint was the comparison of the overall assessment of the models (apicoectomy on 3D-printed model vs. apicoectomy on typodont). The gingiva mask and its suitability for learning surgical incision also had to be evaluated. Additionally, it had to be assessed whether the surgical principles were understood with the help of the models and whether the exercise was recommendable. Due to the exam nature of the exercise, students did not have the opportunity to make a direct comparison between the two models. Since it was known that the exercise was well received in its previous form, the hypotheses to be tested were formulated conservatively, i.e., it was tested whether the printed models were not inferior to the conventional ones.

The ethical approval for this study was obtained from the ethical review committee, Ethikkommission der Ärztekammer Westfalen Lippe und Westfälische Wilhelms Universität Münster, Germany (Ref. No. 2016-675-f-S).

## 3. Results 

### 3.1. Surgical Training Model

Figure 6, Figure 7, Figure 8, Figure 9 and Figure 10 show different stages of the exercise with the 3D-printed surgical training model.

### 3.2. Participants

Group 1 (apicoectomy, *typodont model*) consisted of 35 participants, of whom 21 were female and 14 were male, and who took part in the exercise in the winter semester 2016–2017.

Group 2 (apicoectomy, *3D-printed model*) consisted of 33 participants, of whom 21 were female and 12 were male, and who took part in the exercise in the summer semester 2017.

The examination was carried out as part of a regular course within the scope of the curriculum. There was therefore no voluntary participation for the students.

### 3.3. Evaluation of the 3D-printed Training Model 

#### 3.3.1. Typodont Model

In the overall assessment, the exercise was rated by 35 participants with an average of 9.11 on the VAS (Q1). The question asking whether the exercise contributed to the understanding of apicoectomy (Q2) produced an average rating of 8.94. The question of whether the exercise could be recommended to others was rated with an average of 8.97 (Q3). The average assessment of the gingiva mask for learning incision was 6.05 on the VAS (Q4).

#### 3.3.2. 3D-printed Model

In the overall assessment, the exercise was rated by 33 participants with an average of 9.04 on the VAS (Q1). The question of whether the exercise contributed to the understanding of apicoectomy (Q2) produced an average rating of 9.21. The question asking whether the exercise could be recommended to others was rated with an average of 8.96 (Q3). The average assessment of the gingiva mask for learning incision was 6.44 on the VAS (Q4).

The results are displayed by boxplots (Figure 11).

### 3.4. Free-Text Questions

In the free text questions, 18 participants (54.5%) noted, with respect to the 3D model, that the gingiva mask tears. With the typodont models, 7 participants (20%) criticized that the gingiva mask detached from the model.

### 3.5. Statistical Analysis

Shapiro–Wilk normality tests revealed that, with the exception of Q4, normality cannot be assumed (Table 1). Wilcoxon rank sum tests were therefore carried out to identify differences in the assessments of the two model types. The alternative hypothesis for each test was “The rating for the typodont model is higher than that for the 3D printed”. As the p-values presented in Table 1 reveal, the alternative hypothesis has to be rejected in all cases.

## 4. Discussion

In recent years, a start has already been made to introduce individualized surgical training models [10,11,14] and practice models for conservative or prosthodontic training [13,15] within the framework of student education and postgraduate programs. 

Based on our previous training model [13] we have now developed a 3D-printed surgical training model with a gingival mask. For the first time, individualized 3D-printed training models based on real patient data were used to learn apicoectomy and were compared by students with industrially manufactured typodont models. Thus, the approach of the present study can be assessed as highly innovative and forward-looking. 

It was shown that the 3D-printed surgical training models were not inferior to the industrially manufactured typodont models. Moreover, the presented approach for producing individualized training models offers high flexibility. The 3D-printed models can be redesigned on a regular basis and easily adapted to specific learning goals. Training models for exercising other oral surgical procedures, e.g., wisdom tooth removal or dental implant surgery, can also be made using the presented techniques.

By combining highly precise intraoral scanning [18], computer-aided design and high precision 3D printing [19] realistic models based on real patient data [13] can be created, with the ability to constantly vary the training scenarios. 

With the conventional, off-the-shelf typodont models it was not possible to simulate individual anatomical situations. With the 3D-printed models, which can be produced cost-effectively in small quantities, it is now even feasible to produce a training model for a specific real surgical situation. Such a model could be used by less experienced surgeons to train the surgery before performing it on the patient. This procedure has already been practiced in other disciplines, such as endochonchial interventions [20].

Although high-quality 3D printers are expensive, cost advantages compared to industrially manufactured models can arise even in small order quantities [14]. Material costs for the exchangeable 3D-printed single-use segment for learning apicoectomy were about EUR 10, whereas repair costs of the multiple-use typodont model, which is about EUR 300, were considerably higher. The printed models can also save time, since the repair of a typodont model is time-consuming.

The 3D-printed models presented here for the first time offer students/postgraduates the opportunity to gain practical surgical experience both as part of student training and as part of postgraduate programs. Thus, the practical fundamentals of oral surgery can be learnt first on a model and not, as is usual in practice, on a patient. 

A very important aspect of the present study is that the testing was part of a regular course within the curriculum. There was therefore no voluntary participation in the evaluation of the models. This can clearly be seen as beneficial, as it avoids systematic bias. In a comparable study in the literature [16], the evaluation of the corresponding models took place in the context of a special event with voluntary participants. With such an approach, however, it must be feared that particularly interested or motivated students participate and that the group of students as a whole is not represented. In this case, a systematic bias can be suspected.

A shortcoming of our study is that the exercises were performed by students without surgical experience. As a result, there is a lack of professional evaluation of the models in terms of how well they reflect the reality. Thus, we were not able to check an important quality aspect of the models. Höhne et al. solved this issue in their 2019 study by having their 3D-printed models evaluated, not only by 38 students, but also by 30 experienced dentists [16]. Such a test was missing in our study. On the other hand, however, it must be recognized that Höhne and colleagues refrained from a direct comparison test with a conventional model [16]. This aspect must therefore be considered a strength of the present study. Future studies with experienced surgeons could provide more information about the realism of the 3D-printed models.

There was no direct comparison of the models by the same students, i.e., the typodont models and the 3D-printed models were evaluated by different groups. However, this was necessary in order to avoid distortions in the assessments due to a learning effect.

In our previous study regarding conservative and prosthetic exercises on a 3D-printed model [13], students criticized the lack of a gingiva mask. This time we fabricated a gingiva mask using silicone. From the test student’s perspective, the quality of the gingiva mask for learning surgical incision needs to be improved in both the typodont and 3D-printed models. Therefore, alternative materials (e.g., rubber-like) or technologies (e.g., 3D printing, molding) should be considered for future models.

The missing color difference between anatomical structures was also noted as worthy of improvement in our previous study [13]. Even though we could not establish a color demarcation between tooth and bone, this was not criticized in the surgical training model. Using a material softer than the model material for the demarcation between tooth and bone thus seems to be satisfactory. A further limitation of our models is that there is no differentiation between cortical and cancellous bone structures. This should be taken into account in future models in order to enable an even more realistic simulation of the bony structures.

Werz et al. presented 3D-printed surgical training models fabricated by FDM [14]. These models also featured a gingival mask. In comparison to PolyJet printing, FDM has some technical limitations [6]: the printing resolution is lower, the surface finish rougher and the removal of support structures is much more complex. The support structures and the lower resolution demanded unrealistically thick periodontal ligaments. Furthermore, their gingival mask was thinner and less reproducible. In exchange, their models were much more inexpensive and their gingival mask covered a much larger area.

Due to the developments and progress of the “digital revolution”, many innovations to support new teaching concepts [21] can be expected in dental education. The models presented here are a step in this direction.

## 5. Conclusions 

Individual 3D-printed surgical training models based on real patient data offer a more realistic alternative to industrially manufactured typodont models. For this reason, the authors of the present study are convinced that haptic models will continue to be indispensable in the education, training and examining of dentistry students in the future. However, the gingiva mask for learning surgical incision and flap formation needs further improvement. 

## Figures and Tables

**Figure 1 ijerph-17-02901-f001:**
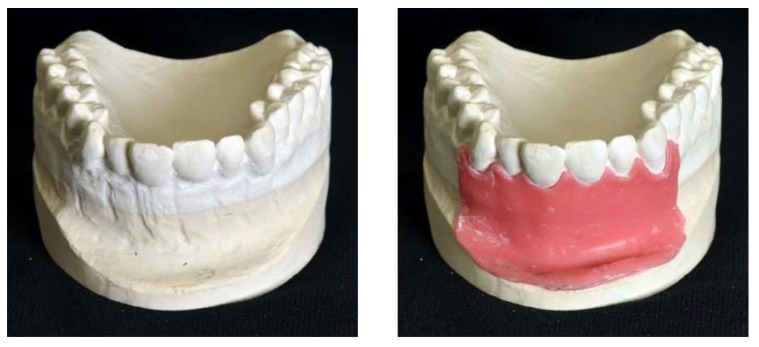
Modified plaster cast (**left**) and modified plaster cast with wax layer (**right**).

**Figure 2 ijerph-17-02901-f002:**
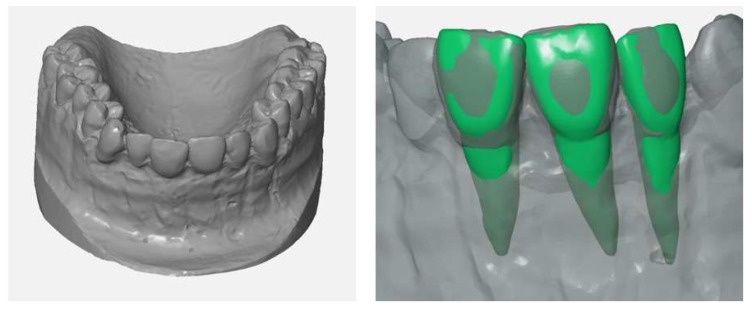
Scanned surface of the plaster cast without wax layer (**left**) and the meshes of the three teeth aligned to the upper jaw (**right**).

**Figure 3 ijerph-17-02901-f003:**
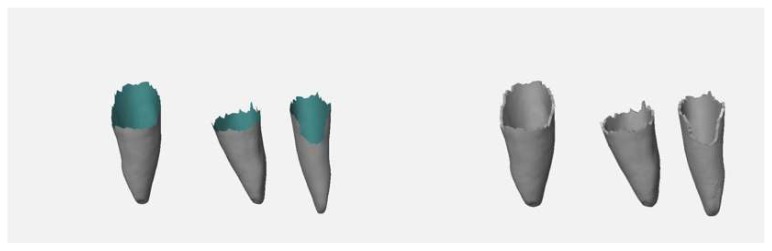
Meshes of the roots (rear faces of the meshes shown in blue-green) (**left**), extruded root surfaces representing the periodontal ligament (**right**).

**Figure 4 ijerph-17-02901-f004:**
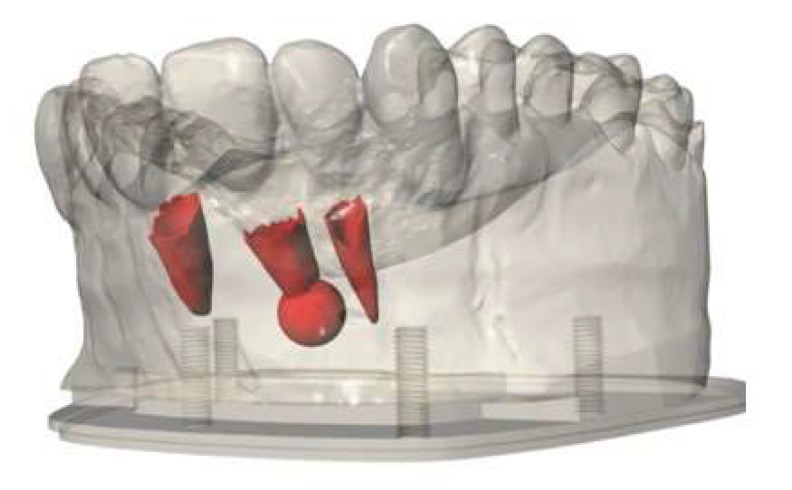
Meshes of the granuloma on tooth 11 and the periodontal ligament on teeth 11, 12 and 21 which will be printed in soft support material (red).

**Figure 5 ijerph-17-02901-f005:**
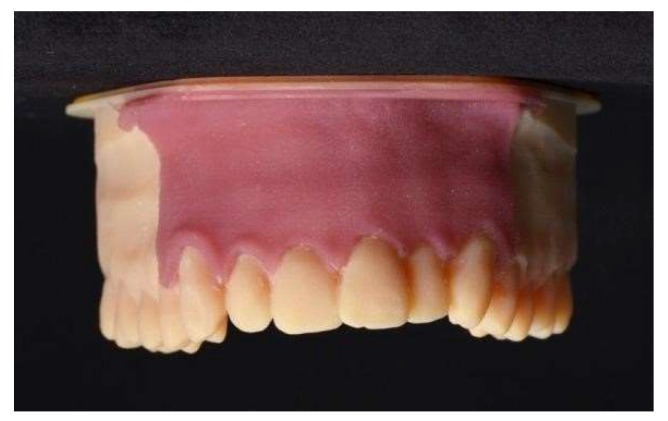
Silicone gingival mask.

**Figure 6 ijerph-17-02901-f006:**
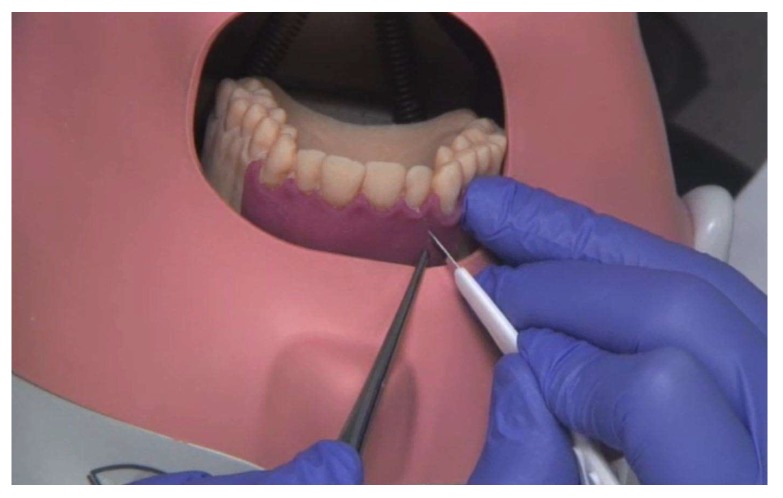
Surgical incision guidance on the 3D-printed model in the phantom.

**Figure 7 ijerph-17-02901-f007:**
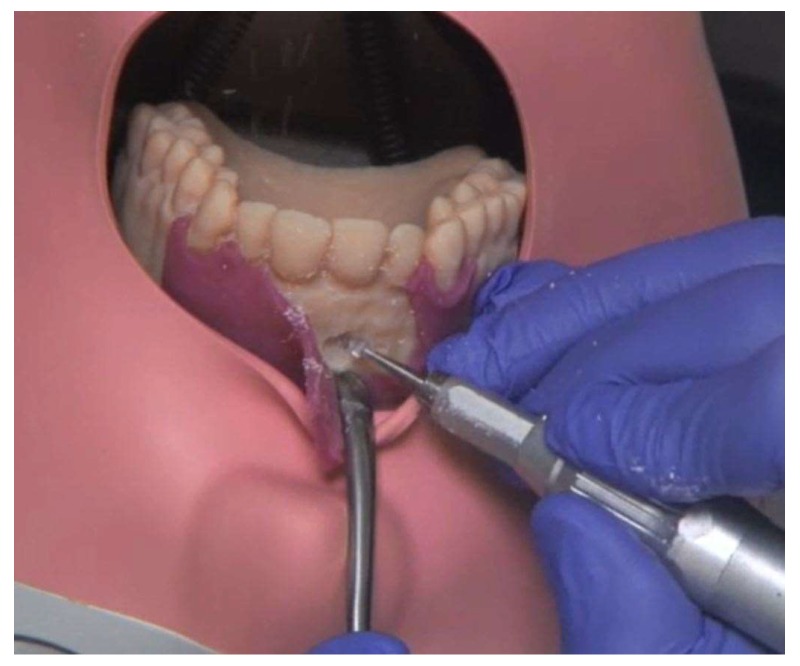
Osteotomy of the root tip.

**Figure 8 ijerph-17-02901-f008:**
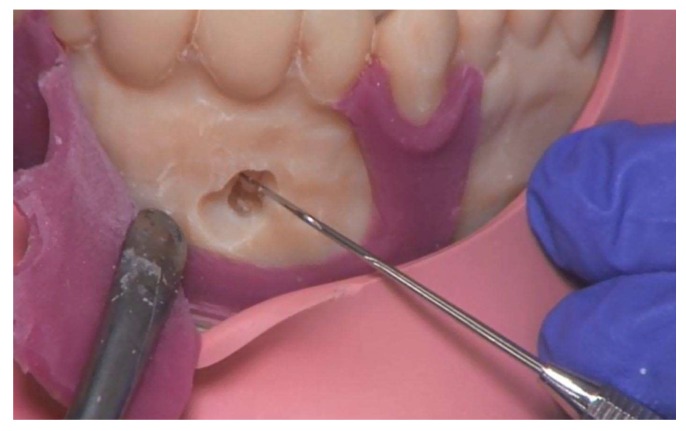
Presentation of the root tip. Note: torn gingiva mask.

**Figure 9 ijerph-17-02901-f009:**
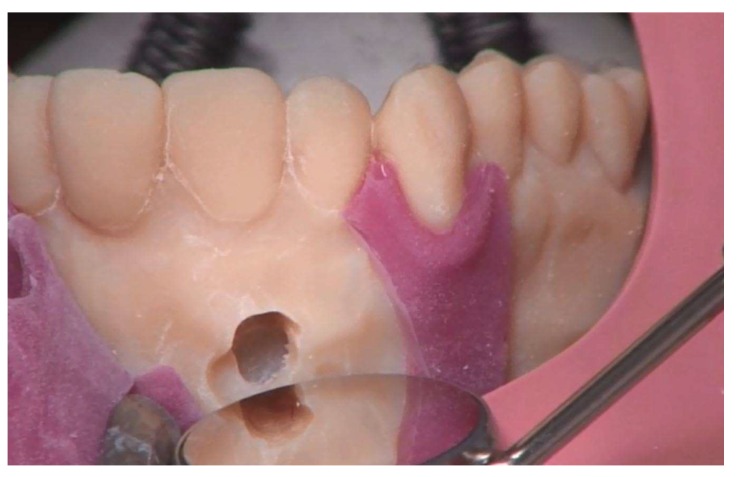
Resected root tip with demarcation to the bone.

**Figure 10 ijerph-17-02901-f010:**
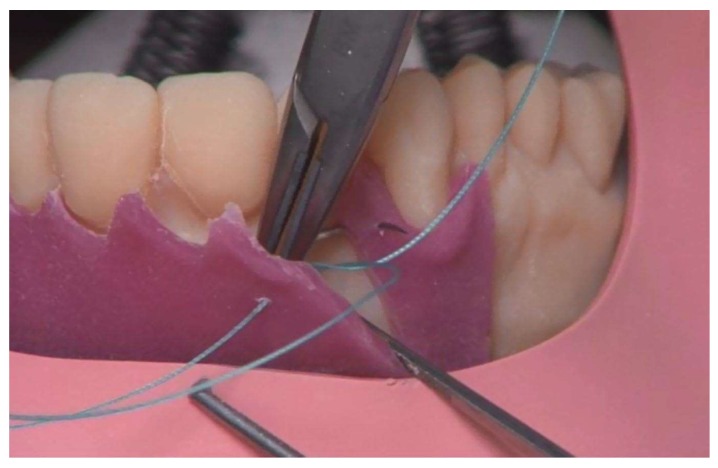
Suture exercise on the gingiva mask.

**Figure 11 ijerph-17-02901-f011:**
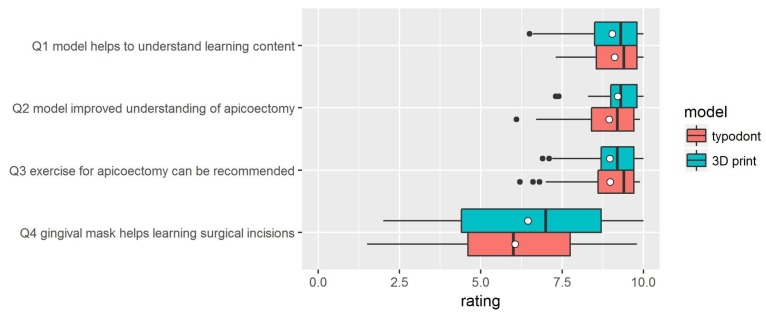
Results of questionnaire. The white dots denote the mean values.

**Table 1 ijerph-17-02901-t001:** Results of Shapiro–Wilk tests for normality of ratings and of Wilcoxon rank sum tests for significant differences between ratings for both model types (alternative hypotheses: true location shift between ratings of typodont model and 3D-printed model is greater than 0).

Question	Shapiro–Wilk Test for Normality; *p*-Value		Wilcoxon Rank Sum Test “Greater Than”
	Typodont	Print	*p*-Value
Q1 model helps to understand learning content	<0.001	<0.001	0.56
Q2 model improved understanding of apicoectomy	<0.001	<0.001	0.87
Q3 exercise for apicoectomy can be recommended	<0.001	<0.001	0.49
Q4 gingival mask helps learning surgical incisions	0.73	0.037	0.76

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
