# Peer review of "3D-printed Surgical Training Model Based on Real Patient Situations for Dental Education"

_ijerph, 2020, doi:10.3390/ijerph17082901_

Round 1

Reviewer 1 Report

Review of the manuscript entitled: “3D Printed Surgical Training Model Based on Real Patient Situations for Dental Education”

The study evaluated the use of 3D printing technology to manufacture individualized surgical training models for root tip resection (apicoectomy) on the basis of real patient data and compared their suitability for dental education against a commercial typodont model by means of questionnaires administered to students of dental school.

The use of new technologies for dental education is interesting.

However, the major concern of the study is that since the final aim of the training is the learning by students of the surgical procedure, the assessment of the skills learned by means of the two tested training models should be done.

Further concerns regards:

Page 2, line 55-58: the hypothesis of the study is not clear. What does the authors expect from students using the printed surgical training model?

In Materials and methods:

Which was the anatomical difference that made the 3D printed model more useful than conventional model?

Were the materials that compose the experimental and conventional model the same? It is not clear if differences in the student evaluation are mainly due to the anatomical features of 3D model (that is closer to the real situation than typodont model) or to the differences in materials and fabrication (i.e. gingival detachment in typodont). This is important since the main advantage of the 3D printed model should be the best simulation of the real situation.

Typodont model are not used in all university for the student training, and the features of the model could not be known by all readers. Please, describe the anatomical features of typodont model and the differences between two models since they are not clear (i.e. presence or not of the periodontal ligament in typodont, presence or not of the granuloma at the apex of the experimental tooth in typodont, length of the tooth…).

Author Response

We would like to thank the editor and the reviewers for their time spent on reviewing our manuscript and their helpful comments. Their suggestions have been implemented in the manuscript. In this letter, we respond point-by-point to the comments of the reviewers and explain the revisions.

As requested by the editor, the language of the manuscript has been revised and the reference format has been modified according to the requirements of the journal. All changes to the manuscript were highlighted using the "Track Changes" function in Microsoft Word.

We hope the manuscript is now suitable for publication in the International Journal of Environmental Research and Public Health.

Reviewer 1:

Comments and Suggestions for Authors

Review of the manuscript entitled: “3D Printed Surgical Training Model Based on Real Patient Situations for Dental Education”

The study evaluated the use of 3D printing technology to manufacture individualized surgical training models for root tip resection (apicoectomy) on the basis of real patient data and compared their suitability for dental education against a commercial typodont model by means of questionnaires administered to students of dental school.

The use of new technologies for dental education is interesting.

However, the major concern of the study is that since the final aim of the training is the learning by students of the surgical procedure, the assessment of the skills learned by means of the two tested training models should be done.

A: The exercise was carried out like an exam. Each student had to show the model to the course assistant after completing the work. The course assistant evaluated the suture, cutting and successful removal of the root tip, but we did not record these evaluations. Our aim was to assess whether those who performed the exercises – the students – accept the 3D printed surgical training model just as well as the typodont model. A remark has been added to the manuscript for clarification (ll. 58-59). However, we will keep this in mind for future studies.

Further concerns regards:

Page 2, line 55-58: the hypothesis of the study is not clear. What does the authors expect from students using the printed surgical training model?

A: We have now reformulated the hypothesis: „Our intention was to evaluate whether dental students accept the 3D printed surgical training model just as well as the popular typodont model.“ (ll. 58-59)

In Materials and methods:

Which was the anatomical difference that made the 3D printed model more useful than conventional model?

A: The 3D printed model also includes the periodontal ligament (represented by a soft material). Remarks regarding the anatomical shortcomings of the conventional model have been added to chapter “2.1. Development of a 3D printed surgical training model” (ll. 64-66).

Were the materials that compose the experimental and conventional model the same?

A: We were not able to obtain exact information on the materials of the typodont model from its manufacturer. We were only told that it is made of an acrylic plastic. Our 3D printed models consist of two photopolymers, one hard (MED690, Stratasys, Rehovot, Israel; tensile strength: 54–65 MPa) and one soft (SUP705, Stratasys), as well as silicone for the gingiva mask (GI-MASK, Coltene®, Liechtenstein) as described in the Materials and Methods sections 2.2 and 2.3 (ll. 102, 121, 123). The name of the soft support material was missing and has been added.

It is not clear if differences in the student evaluation are mainly due to the anatomical features of 3D model (that is closer to the real situation than typodont model) or to the differences in materials and fabrication (i.e. gingival detachment in typodont). This is important since the main advantage of the 3D printed model should be the best simulation of the real situation.

A: A big advantage of the 3D printed model is that it includes the periodontal ligament. This has been clarified in the manuscript (ll. 89-90).

Typodont model are not used in all university for the student training, and the features of the model could not be known by all readers. Please, describe the anatomical features of typodont model and the differences between two models since they are not clear (i.e. presence or not of the periodontal ligament in typodont, presence or not of the granuloma at the apex of the experimental tooth in typodont, length of the tooth…).

A: In the typodont models, the tooth is in direct contact with the hard acrylic, which simulates the jawbone. A delicate periodontal ligament is not present. The apical granuloma, in turn, is simulated in wax. The teeth used are mass-produced and always the same. Anatomical variations such as extremely long or even bent roots cannot be simulated with these models.

We have added this information to the manuscript (ll. 62-66).

Reviewer 2 Report

The topic may be original and appropriate for Int. J. Environ. Res. Public Health.

The standard of written English should be acceptable for a scientific publication and only a few corrections are requested.

INTRODUCTION

-  “Hence, Many authors have developed and presented individualized training models in recent years: For example, around 9 years ago, Güth et al fabricated a typodont implant simulation model and integrated it into dental education”. Please verify the punctuation in this sentence.

- Please avoid to use sentences as this year or last year and substitute them with the year.

- Please improve the aim at the end of this paragraph.

The MATERIALS AND METHODS section quite clearly describes, perhaps it should be reduced.

- No information about the period of recruitment is specified.

- Please, specify if there were some inclusion/exclusion criteria for selecting the sample.

- In the Statistical analysis, a Method Error is not calculated.

The RESULTS addresses the original objective of the study.

In this paragraph, there is no mention of the difference in charges between the model proposed in the study and the one commonly sold. This should also be an important point for discussions.

The DISCUSSION, relates to the findings, is well elaborated and exhaustive.

The CONCLUSIONS briefly exposed the main findings of the study.

REFERENCES are limited to those necessary to support the study.

Author Response

We would like to thank the editor and the reviewers for their time spent on reviewing our manuscript and their helpful comments. Their suggestions have been implemented in the manuscript. In this letter, we respond point-by-point to the comments of the reviewers and explain the revisions.

As requested by the editor, the language of the manuscript has been revised and the reference format has been modified according to the requirements of the journal. All changes to the manuscript were highlighted using the "Track Changes" function in Microsoft Word.

We hope the manuscript is now suitable for publication in the International Journal of Environmental Research and Public Health.

Reviewer 2:

Comments and Suggestions for Authors

The topic may be original and appropriate for Int. J. Environ. Res. Public Health.

The standard of written English should be acceptable for a scientific publication and only a few corrections are requested.

INTRODUCTION

-  “Hence, Many authors have developed and presented individualized training models in recent years: For example, around 9 years ago, Güth et al fabricated a typodont implant simulation model and integrated it into dental education”. Please verify the punctuation in this sentence.

A: We have corrected the punctuation (ll. 41-43).

- Please avoid to use sentences as this year or last year and substitute them with the year.

A: We have corrected this issue (ll. 42-46).

- Please improve the aim at the end of this paragraph.

A: We have revised the aim: “Our intention was to evaluate whether dental students accept the 3D printed surgical training model just as well as the popular typodont model.“ (ll. 58-59).

The MATERIALS AND METHODS section quite clearly describes, perhaps it should be reduced.

A: We have shortened the MATERIALS AND METHODS section.

- No information about the period of recruitment is specified.

A: We have added the missing information about the period of recruitment (summer semester 2016 and winter semester 2016/17 l. 129).

- Please, specify if there were some inclusion/exclusion criteria for selecting the sample.

A: The participants had to be dental students of our university and participants of the operation course 1 (9th semester). There were no further inclusion or exclusion criteria (ll. 127-130).

- In the Statistical analysis, a Method Error is not calculated.

A: A remark concerning the statistical power and the significance level has been added (ll. 143-144)

The RESULTS addresses the original objective of the study.

In this paragraph, there is no mention of the difference in charges between the model proposed in the study and the one commonly sold. This should also be an important point for discussions.

A: We have added information on the costs of the two models, which amount to about 10 € for the 3D print model and about 300 € for the typodont model (ll. 221-223).

The DISCUSSION, relates to the findings, is well elaborated and exhaustive.

The CONCLUSIONS briefly exposed the main findings of the study.

REFERENCES are limited to those necessary to support the study.

Round 2

Reviewer 1 Report

Authors performed some of the required changes to improve the manuscript but not enough for publication.

Author Response

We would like to thank the editor and the reviewers for their time spent on reviewing our manuscript and their helpful comments. Their suggestions have been implemented in the manuscript. In this letter, we respond point-by-point to the comments and explain the revisions.

As requested by the editor, the language of the manuscript has been revised and the reference format has been modified according to the requirements of the journal. All changes to the manuscript were highlighted using the "Track Changes" function in Microsoft Word.

We hope the manuscript is now suitable for publication in the International Journal of Environmental Research and Public Health.

Reviewer 1:

Authors performed some of the required changes to improve the manuscript but not enough for publication.

Answer: We revised the manuscript again, see below.

Academic Editor Notes

The aim of this study was to use 3D printing technology to manufacture individualized surgical training models for root tip resection (apicoectomy) on the basis of real patient data and to 
compare their suitability for dental education against a commercial typodont model.

The study is innovative both for the topic and for the methodology.
The topic is in line with the journal aims.
The purpose of this study is interesting and relevant to the journal. However, there are some parts that should be reorganized and improved as suggested below:

-The Introduction section should be expanded and reinforced; this section is too few to introduce the argument. This section should be one page or a page and a half long.

Answer: We completely revised and extended the introduction and added some new references:

 ll. 42-67:

Digital technology has entered the dental practice on a broad front as it has been proclaimed in the past [1]. By now, certain dental works can be produced in a completely digital workflow [2, 3]. Consequently, it was only a matter of time before the technologies were transferred to dental education. A special focus here is on 3D printing alias additive manufacturing (AM) [4]. 3D printing converts virtual 3D models from computer-aided design (CAD) into haptic objects [5]. In recent years, the development of 3D printing technologies for medical and dental applications has increased significantly. The driving force behind the advancement of 3D printing for these applications is the possibility of creating personalized products, savings on small production runs, easier sharing and easier processing of patient image data [6]. It is expected that these properties of 3D printing will also have great advantages for dental education [6].   

Since in dentistry, similar as in surgery, practical skills are particularly important, the use of training models has been established for years. These models are designed to teach the future dentist practical skills until he or she can apply them to real patients. In dental education, examination situations must also be considered. To maintain fairness, standardized models for conducting examinations should also be thought of. Moreover, models can also be used to visualize special anatomical or pathological situations. This means that (individual) models are of particular importance in dental education, even outside the field of practical training. Last but not least, the models should also promote the students' enthusiasm for education. Good, varied models are therefore a lever for the training of motivated and competent dentists.

In addition to the utilization of haptic models for the training of dentists, the use of virtual reality techniques in training is currently being discussed [7]. However, Nassar and Tekian in (2020) demonstrated in a recent review that an uncritical use of these new technologies is not very promising [8]. The authors of the present study see a great advantage of haptic models in the possibility of learning proprioceptive skills. This aspect has already been emphasized by other authors in the past [9]”.

ll.71-74:

“Furthermore, the ready-made standard models do not usually depict special pathological or anatomic situations. Hence, several authors have developed and presented individualized training models in recent years: for example, Güth et al. fabricated a typodont implant simulation model in 2010 and integrated it into dental education [11].“

ll. 77-80:

“Since then, the rapid advances in 3D printing technologies have allowed more and more educators to produce their own training models [13, 14]. In 2017 our work group presented a 3D printed training model focusing on conservative and prosthetic procedures [15].”

ll. 87-112:

“In a recent study from 2019, Höhne and colleagues [17] used 3D printing to create tooth models in which enamel and dentin could be distinguished. This set these printed models apart from the conventionally fabricated models. The learning effect of preparing a crown was rated as good by the students in this study [17]. Shortly before, Höhne and Schmitter had produced 3D printed models with carious lesions and had them evaluated by students. These models also received a good rating [18].

In summary, there are very promising approaches for the use of 3D printed models in dental education. It is therefore reasonable to assume that this approach should be further pursued and scientifically investigated.

In this contribution, we present an individualized 3D printed surgical training model for root tip resection (apicoectomy), i.e., the removal of an inflammation around the root tip.

In the typodont model used by the students at our university to practice root tip resection (A-J OP OK, Frasaco®, Tettnang, Germany), the teeth are in direct contact with the hard plastic that simulates the jawbone. A sensitive periodontal ligament is not present. The apical granuloma, i.e., an inflammation at the root tip, in turn is simulated in wax. The teeth used are idealized stereotypes. Anatomical variations such as extremely long or even curved roots cannot be simulated with these industrially produced models. Therefore, we have developed a method to create more realistic, individualized training models.

We describe in detail the construction of the new 3D printed surgical training model, which is based on a real patient. It depicts an upper jaw with three anterior root apices, the periodontal ligament around them and an apical granuloma. The material used to represent the periodontal ligament and the apical granuloma is softer than the material used for the other parts of the model. This allows a more realistic representation than in the typodont model. The new model also features a silicone gingival mask for practicing the surgical incision. The mask is manually fabricated directly on the model with a 3D printed matrix. We also present an evaluation of the model by dental students and compare it with their evaluation of the conventional typodont model.”

Added References:

  1. van Noort, R. The future of dental devices is digital. Dent. Mater. Off. Publ. Acad. Dent. Mater. 2012 28, 3–12. https://doi.org/10.1016/j.dental.2011.10.014
  2. Cervino, G., Fiorillo, L., Arzukanyan, A., Spagnuolo, G., Cicciù, M., 2019. Dental Restorative Digital Workflow: Digital Smile Design from Aesthetic to Function. Dent. J. 7, 30. https://doi.org/10.3390/dj7020030
  3. Joda, T., Zarone, F., Ferrari, M.,. The complete digital workflow in fixed prosthodontics: a systematic review. BMC Oral Health 2017, 17. https://doi.org/10.1186/s12903-017-0415-0.
  4. Javaid, M., Haleem, A. Current status and applications of additive manufacturing in dentistry: A literature-based review. J. Oral Biol. Craniofacial Res. 2019,9 , 179–185. https://doi.org/10.1016/j.jobcr.2019.04.004.
  5. Ligon, S.C., Liska, R., Stampfl, J., Gurr, M., Mülhaupt, R. Polymers for 3D Printing and Customized Additive Manufacturing. Chem. Rev. 2017,117, 10212–10290. https://doi.org/10.1021/acs.chemrev.7b0007
  6. Oberoi, G., Nitsch, S., Edelmayer, M., Janjić, K., Müller, A.S., Agis, H. 3D Printing-Encompassing the Facets of Dentistry. Front. Bioeng. Biotechnol. 2018, 6, 172. https://doi.org/10.3389/fbioe.2018.00172
  7. Farronato, M., Maspero, C., Lanteri, V., Fama, A., Ferrati, F., Pettenuzzo, A., Farronato, D. Current state of the art in the use of augmented reality in dentistry: a systematic review of the literature. BMC Oral Health 2019,19. https://doi.org/10.1186/s12903-019-0808-3
  8. Nassar, H.M., Tekian, A. Computer simulation and virtual reality in undergraduate operative and restorative dental education: A critical review. J. Dent. Educ. , 2020, https://doi.org/10.1002/jdd.12138
  9. Torres, K., StaÅ›kiewicz, G., ÅšnieżyÅ„ski, M., Drop, A., Maciejewski, R. Application of rapid prototyping techniques for modelling of anatomical structures in medical training and education. Folia Morphol. 2011, 70, 1–4.

  1. Höhne, C., Schwarzbauer, R., Schmitter, M. 3D Printed Teeth with Enamel and Dentin Layer for Educating Dental Students in Crown Preparation. J. Dent. Educ. 2019,83, 1457–1463. https://doi.org/10.21815/JDE.019.146
  2. Höhne, C., Schmitter, M. 3D Printed Teeth for the Preclinical Education of Dental Students. J. Dent. Educ. 2019, 83, 1100–1106. https://doi.org/10.21815/JDE.019.103
  3.  
  4.  
  5. Zhao, Z.; Xu, S.; Wood, B.J.; Tse, Z.T.H. 3D Printing Endobronchial Models for Surgical Training and Simulation. J. Imaging 2018, 4, 135

-The name of the statistician who led the analysis should also be indicated and must be specified if he was independent.

Answer: The sample size calculation and statistical set up was done by Dr. Raphael Koch (Institute of Biostatistics and Clinical Research, University of Münster). Statistical analysis was done by D.D. We have added this information to the manuscript (ll. 185-187).

-The Authors should motivate and better explain the advantages of this technique and the limitations of the study.

Answer: We have added additional information on the advantages of the proposed technique and on the limitations of our study to the introduction and discussion sections.

ll. 265-278:

“By combining highly precise intraoral scanning [20], computer-aided design and high precision 3D printing [21] realistic models based on real patient data [15] can be created, with the ability to constantly vary the training scenarios.

With the conventional, off-the-shelf typodont models it was not possible to simulate individual anatomical situations. With the 3D-printed models, which can be produced cost-effectively in small quantities, it is now even feasible to produce a training model for a specific real surgical situation. Such a model could be used by less experienced surgeon to train the surgery before performing it on the patient. This procedure has already been practiced in other disciplines, such as endochonchial interventions. [22].

Although high-quality 3D printers are expensive, cost advantages compared to industrially manufactured models can arise even in small order quantities [16]. Material costs for the exchangeable 3D printed single-use segment for learning apicoectomy were about 10 €, whereas repair costs of the multiple-use typodont model, which is about 300 €, were considerably higher. The printed models can also save time, since the repair of a typodont model is time-consuming.”

ll. 282-301:

“A very important aspect of the present study is that the testing was part of a regular course within the curriculum. There was therefore no voluntary participation in the evaluation of the models. This can clearly be seen as beneficial, as it avoids systematic bias. In a comparable study in the literature [18], the evaluation of the corresponding models took place in the context of a special event with voluntary participants. With such an approach, however, it must be feared that particularly interested or motivated students participate and that the entire group of students is not represented. In this case, a systematic bias can be suspected.

A shortcoming of our study is that the exercises were performed by students without surgical experience. As a result, there is also a lack of professional evaluation of the models in terms of how well they reflect the reality. Thus, we were not able to check an important quality aspect of the models. Höhne et al. solved this issue in their 2019 study by having their 3D printed models evaluated not only by 38 students but also by 30 experienced dentists [18]. Such a test was missing in our study. On the other hand, however, it must be recognized that Höhne and colleagues refrained from a direct comparison test with a conventional model [18]. This aspect must therefore be considered a strength of the present study. Future studies with experienced surgeons could provide more information about the realism of the 3D printed models. There was no direct comparison of the models by the same students, i.e, the typodont models and the 3D printed models were evaluated by different groups. However, this was necessary in order to avoid distortions in the assessments due to a learning effect.”

ll. 312-317:

“The missing color difference between anatomical structures was also noted as worthy of improvement in our previous study [51]. Even though we could not establish a color demarcation between tooth and bone, this was not criticized in the surgical training model. Using a material softer than the model material for the demarcation between tooth and bone thus seems to be satisfactory. A further limitation of factor our models is that there is no differentiation between cortical and cancellous bone structures. This should be taken into account in future models in order to enable an even more realistic simulation of the bony structures. Werz et al. presented 3D printed surgical training models fabricated by FDM [16]. These models also featured a gingival mask.”

- Discussions section should be expanded. The reference list is up to date but 10 references are few to clarify the topic. The reference list should be increased

Answer: The discussion was expanded (see above) and some new references were added (see above).

Data reported in the Methods section are appropriate and precisely described; they appear to be reproducible.

Results are reported clearly and adequately supported by images. The images are clear and indicative of the content.
According to this Reviewer’s consideration, publication of the present manuscript is recommended after minor revision